# Towards Personalized Medicine: Non-Coding RNAs and Endometrial Cancer

**DOI:** 10.3390/healthcare9080965

**Published:** 2021-07-30

**Authors:** Anna Franca Cavaliere, Federica Perelli, Simona Zaami, Roberto Piergentili, Alberto Mattei, Giuseppe Vizzielli, Giovanni Scambia, Gianluca Straface, Stefano Restaino, Fabrizio Signore

**Affiliations:** 1Azienda USL Toscana Centro, Gynecology and Obstetric Department, Santo Stefano Hospital, 59100 Prato, Italy; annafranca.cavaliere@uslcentro.toscana.it; 2Azienda USL Toscana Centro, Gynecology and Obstetric Department, Santa Maria Annunziata Hospital, 50012 Florence, Italy; alberto.mattei@uslcentro.toscana.it; 3Department of Anatomical, Histological, Forensic and Orthopedic Sciences, Sapienza University of Rome, Viale Regina Elena 336, 00161 Roma, Italy; simona.zaami@uniroma1.it; 4Institute of Molecular Biology and Pathology, Italian National Research Council (CNR-IBPM), 00185 Rome, Italy; roberto.piergentili@cnr.it; 5Gynecologic Oncology Unit, Fondazione Policlinico Universitario A. Gemelli IRCCS, 00168 Rome, Italy; giuseppevizzielli@yahoo.it (G.V.); giovanni.scambia@policlinicogemelli.it (G.S.); 6Obstetrics, Gynecology and Pediatrics Department, Udine University Hospital, DAME, 33100 Udine, Italy; Stefano.restaino@asufc.sanita.fvg.it; 7Division of Perinatal Medicine, Policlinico Abano Terme, 35031 Abano Terme, Italy; glstraface@gmail.com; 8Obstetrics and Gynecology Department, USL Roma2, Sant’Eugenio Hospital, 00144 Rome, Italy; fabrizio.signore@aslroma2.it

**Keywords:** endometrial cancer, molecular biology, non-coding RNA, competing endogenous RNA, biomarkers, prognostic factors, risk factors

## Abstract

Endometrial cancer (EC) is the most frequent female cancer associated with excellent prognosis if diagnosed at an early stage. The risk factors on which clinical staging is based are constantly updated and genetic and epigenetic characteristics have recently been emerging as prognostic markers. The evidence shows that non-coding RNAs (ncRNAs) play a fundamental role in various biological processes associated with the pathogenesis of EC and many of them also have a prognosis prediction function, of remarkable importance in defining the therapeutic and surveillance path of EC patients. Personalized medicine focuses on the continuous updating of risk factors that are identifiable early during the EC staging to tailor treatments to patients. This review aims to show a summary of the current classification systems and to encourage the integration of various risk factors, introducing the prognostic role of non-coding RNAs, to avoid aggressive therapies where not necessary and to treat and strictly monitor subjects at greater risk of relapse.

## 1. Introduction

Endometrial cancer (EC) has been managed according to different classification systems based on several risk factors characteristic of each patient. Over the years, new risk factors related to the EC prognosis have been discovered and a better characterization of the disease has allowed a more precise treatment. 

The first classification was mainly based on pathological aspects, from which endocrine, metabolic and clinical behavior derived. It was called the Bokhman’s dualistic theory, according to which EC has been traditionally classified into the following two main groups: type I and type II [1].

The European Society for Medical Oncology (ESMO), the European Society of Gynecological Oncology (ESGO) and the European Society for Radiotherapy and Oncology (ESTRO) published a consensus in 2016 that defined six risk categories (low, intermediate, high-intermediate, high, advanced, metastatic) to better guide adjuvant therapy after staging surgery [2].

The Cancer Genome Atlas (TCGA) described for the first time in 2013 the following four distinct endometrial cancer subgroups: polymerase epsilon (POLE) mutated, hypermutated secondary to microsatellite instability (MSI), low copy number and high copy number, based on molecular features [3].

The Randomized phase III Trial of Molecular Profile based versus Standard Adjuvant Radiotherapy in Endometrial Cancer (PORTEC-4a) is an ongoing prospective, multicenter, randomized trial and is enrolling women with stage I EC classified as being within the high-intermediate prognostic group according to the ESMO/ESGO/ESTRO consensus. This subset of patients will be submitted to external beam radiation (EBRT), vaginal brachytherapy or no adjuvant treatment based on its molecular risk profile. The trial’s aim is to compare the outcomes of patients who receive adjuvant vaginal brachytherapy, which is the current standard for high-intermediate risk EC, with EBRT or only follow up in cases of unfavorable or favorable molecular risk profile, respectively (NCT03469674) [4,5].

Recently, the ESGO, the ESTRO and the European Society of Pathology (ESP) updated an evidence-based guideline integrating clinical factors from the 2016 ESMO/ESGO/ESTRO with molecular factors from the TCGA molecular-based classification [6].

Non-coding RNAs (ncRNAs) are defined as transcripts of nucleotides with little or no protein-coding capacity. They have been associated with various malignant tumors as their deregulation can lead to changes in the function of oncogenes or tumor suppressors, showing their role in tumorigenesis. Their gene expression regulation can occur in different steps, at epigenetic, transcriptional, posttranscriptional and other stages [7]. Some deregulated ncRNAs have recently been emerging among the risk factors that can better define the EC behavior [8].

This review aims to summarize the clinical and biological risk factors on which the current classification systems are based and to include the ncRNAs among the prognostic factors to tailor treatments and oncological surveillance of each patient.

## 2. Endometrial Cancer Overview

EC is the most frequent female cancer in developed countries. 

It affects about seventy-nine out of 100,000 women in Europe with a median age at diagnosis of sixty-two years [9,10].

Several risk factors have been defined and grouped in metabolic alterations, such as obesity, diabetes, polycystic ovary syndrome (PCOS), genetic predisposition, such as Lynch Syndrome and Cowden Syndrome and other conditions such as tamoxifen use and infertility [11,12].

Due to the increase in the incidence of obesity worldwide, the incidence of EC is also significantly increasing [13].

The prognosis of EC patients seems to be closely linked to the stage of the disease at diagnosis. 

An early stage EC patients’ prognosis is generally favorable [9]. Currently, the treatment of patients with initial disease is based on EC risk factors: low risk patients are treated with surgery alone or followed by brachytherapy and/or EBRT, combined with platinum-based chemotherapy in stage I high risk and in stage II patients [2]. 

Advanced-stage EC patients show a higher risk of pelvic or distant recurrence, the greater for the non-endometrioid tumor histology [9]. For advanced-stage disease patients belonging to the high-risk group (i.e., stage III endometrioid tumor without residual disease after surgery, appropriately staged) the treatment involves EBRT, which represents the current standard, combined with chemotherapy in the context of clinical trials. Finally, for advanced or metastatic EC patients, surgery should only be considered to obtain a complete cytoreduction with no residual disease, and a systemic palliative treatment, such as carboplatin and paclitaxel-based chemotherapy or hormonal therapy, should be offered [2].

The 5-year overall survival (OS) is 95% for stage I women and 69% for stage II [9]. However, about 13% of high-risk patients and 3% of women classified as low risk have a relapse [14]. The prognosis of patients affected by recurrent or advanced disease (stage III or IV) is poor, with 5-year OS rates related to metastatic disease ranging from 15 to 17% [9].

Using that risk stratification based on the current classification systems, a percentage of patients is still overtreated or undertreated, up to 10% of early stage patients will experience distant metastasis and 7 high-intermediate risk patients need to undergo vaginal brachytherapy to prevent a recurrence [5].

For this reason, the researchers’ efforts are currently focused on the genetic and epigenetic fields, in order to find unknown risk factors to better define EC patients’ prognosis and management.

## 3. Classification Systems of Endometrial Cancer and Related Risk Factors

### 3.1. Dualistic Model

In 1981, Bokhman described the hypothesis that EC could be grouped into two categories that explained the different tumor behavior, especially linked to the prognosis of patients affected by the different types. He defined the so-called “two pathogenetic types” based on women’s signs and the tumor’s peculiarities and reported their influence on grading and the patient’s survival [1].

Type I EC is generally associated with specific clinical and pathological features such as low grade (G1 or G2), rare LVSI, frequent prevalence of superficial myometrial invasion, long duration of symptoms that make an early diagnosis of the disease easier and favorable prognosis.

Type II EC is associated with high grade (G3), frequent LVSI, frequent prevalence of deep myometrial invasion, short duration or total absence of symptoms and doubtful prognosis.

### 3.2. ESMO/ESGO/ESTRO Classification

In 2016, Colombo et al., in the context of the ESMO/ESGO/ESTRO consensus on endometrial cancer and its diagnosis, treatment and follow-up, defined six risk groups for adjuvant therapy. The authors presented a classification system based on the following several prognostic factors: age, FIGO stage, depth of myometrial invasion, tumor differentiation grade, tumor histotype (endometrioid versus serous and clear cell) and LVSI [2].

-Age.-FIGO stage.According to the FIGO (International Federation of Gynecology and Obstetrics) EC is classified in four stages based on the following three elements: the size of the tumor (T), the loco-regional lymph nodes involvement (N) and the spread to distant sites (M) [15] (Table 1).

-Depth of myometrial invasion.-Tumor differentiation grade.-Tumor histotype.-LVSI.

The above listed risk factors are grouped in six risk categories according to Colombo et al. [2], as reported in Table 2.

### 3.3. TCGA Classification

TCGA stands for The Cancer Genome Atlas and is a project aimed at creating a catalogue of the genetic mutations responsible for the various types of cancer. In 2013, the Cancer Genome Atlas Research Network contributed to the genomic characterization of EC. The authors performed an integrated genomic and proteomic analysis of 373 endometrial tumors and provided a diagnostic classification based on molecular biology [3].

They thus classified EC in four prognostic groups. Their molecular characterization data showed that approximately one quarter of endometrioid histotype and G3-graded tumors have a molecular phenotype similar to serous EC [16]. Since patients with endometrioid EC generally have a good clinical course, while those with serous histotype usually have a poor prognosis, the authors concluded that the molecular classification can lead to an improved management of these patients, assuming the tendency to treat tumors with specific and unfavorable genetic changes with chemotherapy instead of radiotherapy alone [3].

The authors reported four prognostic categories as follows:(1)POLE ultramutated

Patients belonging to this subgroup show the best prognosis and the longer progression-free survival.

The most frequent histotype associated with this genetic feature is endometrioid.

Some authors showed an association between POLE ultramutated and specific genes alterations, as follows: POLE, PTEN, PIK3R1, PIK3CA, FBXW7, KRAS, TP53 [17].

The POLE gene encodes the catalytic subunit A of DNA polymerase epsilon, which is involved in DNA replication and repair [14]. In EC patients, the most common mutations detected in POLE were P286R and S297F in exon nine and V411L, L424V and L424I in exon thirteen [18].

(2)Microsatellite instability hypermutated

Patients belonging to this subgroup show intermediate prognosis.

The most frequent histotype associated with this genetic feature is endometrioid.

Some authors showed an association between microsatellite instability hypermutated and specific genes alterations, as follows: PTEN, KRAS, ARID1A [17].

MSI represents the phenotypic evidence that DNA mismatch repair (MMR) is not functioning normally. MMR deficiency is linked to sporadic and inherited cancers of the colon, endometrium and brain. Biallelic inactivation of one of four

MMR genes (MLH1, MSH2, MSH6 or PMS2) results in an accumulation of mutations, the so-called MSI. The loss of MMR genes can occur through a combination of the following mechanisms: point mutations, small insertions or deletions, copy number changes, loss of heterozygosity, structural rearrangements and methylation of a gene promoter [19].

(3)Low copy-number

Patients belonging to this subgroup show intermediate prognosis.

The most frequent histotype associated with this genetic feature is endometrioid.

Some authors showed an association between a low copy-number and specific genes alterations, as follows: CTNNB1, PTEN [17].

Copy number changes are comprised in the genome structural variation: they consist in duplication or deletion events that affect a considerable number of DNA base pairs.

(4)High copy-number

Patients belonging to this subgroup show the worst prognosis.

The most frequent histotype associated with this genetic feature is serous.

Some authors showed an association between a high copy-number and specific genes alterations, as follows: TP53, FBXW7, PPP2R1A [17].

### 3.4. PORTEC-4a Classification

The PORTEC-4a is the first clinical trial applying molecular-integrated risk profiles in EC patients. It is a randomized trial of molecular-based versus standard recommendations to determine adjuvant treatment in women with early stage endometrial cancer.

Patients will be followed for vaginal recurrence after surgery if classified as “favorable molecular risk profile”. They will be treated with vaginal brachytherapy if classified as “intermediate molecular risk profile”. They will be treated with EBRT if classified as “unfavorable molecular risk profile”. The primary endpoint is the vaginal recurrence, while the secondary endpoints are the recurrence-free and overall survival, the recurrence location, the adverse events incidence, the delayed symptoms and quality of life and EC-related healthcare costs [4].

The authors reported three prognostic categories as follows:(1)Favorable:
-POLE mutation,-or NSMP (No Specific Molecular Profile),-without CTNNB1 mutations.
(2)Intermediate:
-Mismatch repair-deficient (MMRd),-or NSMP,-with CTNNB1 mutations.
(3)Unfavorable:
-substantial LVSI,-TP53 abnormal immunohistochemical staining-or L1CAM overexpression.


The CTNNB1 mutation causes the activation of the Wnt signaling pathway and has been shown to be associated with tumorigenesis in many types of human cancers. This gene encodes beta-catenin protein, which is involved in the Wnt signaling pathway and regulates cell growth, motility and differentiation [20]. Its mutation is associated with EC favorable prognosis.

P53 gene encodes for the tumor suppressor p53 protein (TP53). Its abnormal immunohistochemical staining is associated with poor EC prognosis: the loss of tumor suppressor p53 would result in a high degree of genomic instability and rapid tumor progression and invasion. The immunohistochemical staining abnormality reflects, as opposed to normal p53 wild-type staining, a mutational status of TP53. TP53 alteration causes the loss of genomic stability, DNA repair capacity, cellular senescence and apoptosis; it is one of the most frequent abnormalities in human cancer [21].

L1CAM is a 200 to 220kDa transmembrane protein of the immunoglobulin family, which may promote aggressive tumor behavior by driving cell proliferation, migration, invasion and metastasis [22]. It is associated with more aggressive EC hystologies, LVSI, advanced stage disease and more distant relapses [23].

## 4. Epigenetic Risk Factors of EC

The term epigenetics refers to heritable phenotype changes that do not involve alterations in the DNA sequence.

Epigenetic mechanisms consist of DNA methylation, histone modifications, nucleosome remodeling and modulation of the chromatin structure [24].

Human genomes have been shown to be divided into the following two groups: protein-coding messenger RNAs (mRNAs) and RNAs without coding potential, also known as noncoding RNAs (ncRNAs).

A non-protein-coding genome represents most of the whole mammalian genome.

Several authors demonstrated epigenetic signatures to be characteristic of many cancers.

EC seems to be associated with gene function alterations mediated by ncRNA that may control cell mobility and invasion (important for metastasis formation), angiogenesis, resistance to chemotherapeutic agents and gene transcriptional regulation.

### 4.1. ncRNA Role in EC Pathogenesis

Non-coding RNAs (ncRNAs) are involved in many cellular processes and are associated with the tumorigenesis of several human cancers, showing abnormal expression patterns in many tumoral tissues. Some authors reported ncRNAs’ deregulation as being responsible for the genesis and progression of various tumors and proposed their use as biomarkers for that disease [7].

As protein expression in EC seems to be altered, the protein coding regulation anomalies typically associated with RNA inhibitory activity have often been suggested [25,26,27,28,29,30].

NcRNAs do not have a coding protein capacity, but they can regulate gene expression and, acting in an oncogenic or tumor suppressor sense, they may promote various phases of tumorigenesis.

They are classified, according to their length, and defined as “long” ncRNAs (lncRNAs) in case of the presence of over 200 nucleotides, as “short” ncRNAs (sncRNAs) when they are made up of less than 200 nucleotides [7].

A subgroup of sncRNAs is represented by “micro” RNAs (miRNAs) that are usually composed of about 20 nucleotides.

Their role is to bind the target mRNA by silencing it, resulting in a reduction in that target mRNA expression, thus reducing the production of tumor suppressor proteins or indirectly increasing that of oncogene ones, contributing to the pathogenesis of several cancers.

### 4.2. ncRNA as Risk Factor for EC

The ncRNAs can be classified in long and short ncRNAs and they can also be identified as competing endogenous RNAs (ceRNAs), as they can “compete” with the miRNAs binding them and restoring the coding capacity of the target mRNAs, free of the silencing effect of the link with the miRNAs [31].

The interaction between ceRNAs, miRNAs and mRNAs plays an important role in many cellular processes, including proliferation, apoptosis, angiogenesis, migration and invasion (formation of metastases) and cell cycle transition regulation [32].

This interaction occurs between homologous sequences, with an lncRNA that inhibits the binding of one miRNA to its mRNA target. This kind of interaction, and its deregulation, represents the basis of various types of cancer [33,34].

LncRNA and sncRNA are always expressed in the opposite sense, i.e., if the lncRNA is up-regulated (over-expressed), the corresponding miRNA is down-regulated, because more lncRNA molecules can bind the miRNA and the target mRNA expression is not reduced. Therefore, if the target mRNA encodes an oncogene, the up-regulation of the lncRNA makes it an oncogene (lncRNA and mRNA change together, increasing or decreasing at the same time), while the corresponding miRNA is functionally a tumor suppressor (inverse sign). The opposite mechanism is still valid.

Table 3 and Table 4 show, respectively, the ncRNAs that play a role as an oncogene or a tumor suppressor in EC pathogenesis and for which a functional characterization is available (Table 3 and Table 4).

The so-called “ceRNAs network” has been the subject of some studies that show the link between the ceRNAs alteration and some cancer characteristics and natural history, including EC [35].

The most frequent mechanisms related to EC tumorigenesis deriving from the deregulation of the ceRNA network are uncontrolled cell growth, apoptosis alterations, deregulation of cell invasion and migration, drug resistance development, epithelial to mesenchymal transition (Table 3 and Table 4).

Another great potential of ncRNAs identified by some authors, lies in the possibility of using these molecules as therapeutic targets for personalized medicine [36].

This would minimize the toxic effects of systemic therapies allowing a different targeted therapy for each subgroup of patients.

### 4.3. ncRNA Selected for EC Patients Risk Stratification

It is surprising to note that the association between ncRNA and EC has only recently been emerging in the literature, and that most of the papers regarding this association have been concentrated into the last three years.

A search of the PubMed database was carried out by two researchers independently and the following search strings were used: “endometrial cancer AND non coding RNA AND Kaplan–Meier” and “endometrial cancer AND competing endogenous RNA AND survival”. Only papers related to the issue of the prognostic role of ceRNA in the context of EC and published from January 2018 to January 2021 were considered. The first search string retrieved 12 studies, including 20 ceRNAs correlated with the survival data of the patients under study. The second search string showed 11 studies regarding the prognostic role of 19 ceRNAs (a five-ceRNAs study derived from both search strings). Moreover, six ceRNAs were the subject of several selected studies; therefore, a total of 30 molecules were identified and considered for a prognostic stratification of EC patients.

Some ceRNAs, such as AC074212.1, ADARB2-AS1, C2orf48, C8orf49, C10orf91, FER1L4, FP671120.4, GLIS3-AS1, HOXB-AS1, LINC00483, LINC00491, LINC01143, LINC01352, LINC01410, LINC02381, MIR503HG, PCAT1, RP11-357H14.17 and RP11 89K21.1 seem to be associated with poor prognosis, while AL596188.1, KCNMB2-AS1, LA16c-313D11.11, LINC00237, LINC00475, LINC00958 and LNCTAM34A show a positive prognostic effect. Two ceRNAs, AC110491.1 and LINC00261, showed conflicting data, exhibiting an opposite prognosis in the different studies that analyzed them. 

All the above-mentioned deregulations in ncRNAs expression were identified comparing EC tissue with normal endometrial tissue, similarly to what was performed for the collection of TCGA data.

Figure 1 shows the risk factors that should be considered for EC patients’ stratifications to guarantee them an accurate and broad prognostic characterization, which takes into account both the established and emerging risk factors. They have been grouped into the following four classes: clinical, pathological, genetic and epigenetic prognostic factors. The clinical consolidated risk factor is represented by age at diagnosis; it assumes a negative prognostic value in case of older age. Among the pathological factors a higher FIGO stage and higher grading are related to the worst outcome, as for non-endometrioid hystotype and the presence of LVSI. The genetic alterations to consider for EC patients’ stratification, derived from emerging evidence, are related to the following genes: POLE, MSI-related genes (MLH1, MSH2, MSH6 or PMS2), copy number changes, CTNNB1, TP53 and L1CAM. Finally, the epigenetic risk factors to consider in a wide stratification model should include those ceRNAs related to EC patients’ survival outcome, as follows: AC074212.1, AC110491.1, ADARB2-AS1, AL596188.1, C2orf48, C8orf49, C10orf91, FER1L4, FP671120.4, GLIS3-AS1, HOXB-AS1, KCNMB2-AS1, LA16c-313D11.11, LINC00261, LINC00237, LINC00475, LINC00483, LINC00491, LINC00958, LINC01143, LINC01352, LINC01410, LINC02381, LNCTAM34A, MIR503HG, PCAT1, RP11-357H14.17 and RP11-89K21.1.

In the epigenetic risk factors box, the ncRNAs are listed in alphabetical order. The data in the figure are retrieved from references [2,3,4,26,30,35,36,38,63,65,77,82,84,95,96,97].

## 5. Discussion

EC risk factors predicting a poor prognosis and ensuring an adequate treatment are currently being studied. Nowadays, research is increasingly oriented towards discovering genetic risk factors that can identify high-risk EC quickly and with non-invasive techniques. In this context, TCGA work has represented an epochal turning point for cancer patients’ personalized management, allowing it to be selected according to genetic and epigenetic characteristics [3].

Epigenetic modifications are gaining increasing importance for the characterization of many diseases including cancer and a group of molecules is emerging as identifiable risk factors that help to establish clinical prognosis: the ncRNAs.

There is a correlation between some ncRNAs alterations and the predictive course of EC, representing the possibility of including these molecules in stratifying patients at greater risk of relapse and a poor outcome [8].

The current literature shows an upregulation or a downregulation of some types of ncRNAs in patients with EC compared to controls. 

In particular, some studies have shown a correlation between EC patients’ survival and the presence of specifically deregulated ncRNAs (Table 5).

In this review, we have selected the papers that demonstrated an association between EC patients’ survival and the deregulation of specific ncRNAs.

The enormous value of the molecular finding lies in selecting patients who show a typical mapping of ncRNAs, making it possible to define the prognostic category and the most appropriate therapeutic and surveillance plan.

Furthermore, these molecules may represent diagnostic markers and therapeutic targets.

Therefore, it is evident that the old, dualistic classification of EC as type I and II [1] is no longer appropriate, and even that based on coding gene mutations is amply insufficient for the goal of personalized medicine.

The integrated classification strength is that ncRNAs can become both diagnostic, prognostic and therapeutic markers and for the first time the building of an “epigenetic profile” could assume a role in the treatment choice.

The new therapeutic strategy would consider the entire metabolic pathway of the identified altered molecules.

The weak point consists of the fact that, due to the multiple interactions between ceRNA–miRNA–mRNA, more targets are hit simultaneously, directly or indirectly. Therefore, it becomes difficult to look for a single modified protein to identify and treat EC effectively because some metabolic pathways are altered in several steps. A complete analysis of several molecules simultaneously would be appropriate to obtain a tailored medicine. Each patient is characterized by a specific set of molecular alterations, whose target is well defined, and that establishes the best therapeutic strategy. Making a selection of the available molecules with a prognostic value will be appropriate and will have to be the subject of future prospective studies. A possible limitation to the use of ncRNA as prognostic markers in clinical practice is represented by their tissue detection using quantitative real-time PCR, which requires the use of expensive fluorescently labeled probes.

Studying ncRNA expression based on current classifications to identify the correspondence between anatomical, genetic and epigenetic characteristics is a challenging goal. 

Wang et al. demonstrated the superiority of the prognostic accuracy of multiRNA combined with clinical-stage over clinical stage alone, according to FIGO and over some genetic prognostic predictive models available in the literature [95].

Some recent reviews had already highlighted the role of ncRNAs in developing female cancers [8,98,99], but they mainly focused on their mechanism in tumorigenesis. The present review is the first to identify and select a pool of ncRNAs that have a prognostic role and can influence the therapeutic choice based on the epigenetic profile of EC patients.

Finding a classification system that includes the integration of all the known risk factors would be a step towards truly tailored medicine. NcRNAs can be expressed in cancerous tissue but are also detectable in the blood circulation; therefore, they can be considered biomarkers. This would allow for not only their role in the EC prognostic classification but also in a future early diagnostic panel for several neoplasms. Moreover, the deregulation of the ncRNA correlating to the survival of EC patients selected by some studies (see Figure 1, epigenetic risk factors) seem to be epigenetic prognostic factors independent of the other currently established clinical risk factors.

The ongoing studies on genetic panels and integration between clinical and genetic factors will produce valuable data to identify more accurate oncological pathways for EC patients [4]. However, it is mandatory to also consider the epigenetic aspects of a tumor, detectable in the first phase by extraction from the tumor tissue, in a second phase from the circulating blood of EC patients. Figure 1 summarizes the old and new risk factors to be used for an optimal prognostic classification. For a complete risk stratification, it would be appropriate to create prospective data collections considering all the characteristics shown in Figure 1. In the hypothesis supported by some studies that clinical, genetic and epigenetic risk factors play an independent role, it would be necessary to study the prognostic accuracy of each prognostic factor taken individually and integrated with the others through multivariate analyses and logistic regressions.

It would be appropriate to collect all the reported risk factor information in a future observational and interventional study to obtain a global clinical and molecular characterization of EC, which would allow for the comparison of the various studies not only in terms of clinical outcomes but also regarding the molecular biology underlying tumor transformation and its diagnostic and therapeutic potential.

## 6. Conclusions

Therapeutic pathways tailored to the genetic and epigenetic characteristics of EC patients are the basis of precision medicine in oncology. Indeed, it could reduce the side effects of unnecessary therapies for patients without negative prognostic factors, while adequately treating high-risk EC patients.

## Figures and Tables

**Figure 1 healthcare-09-00965-f001:**
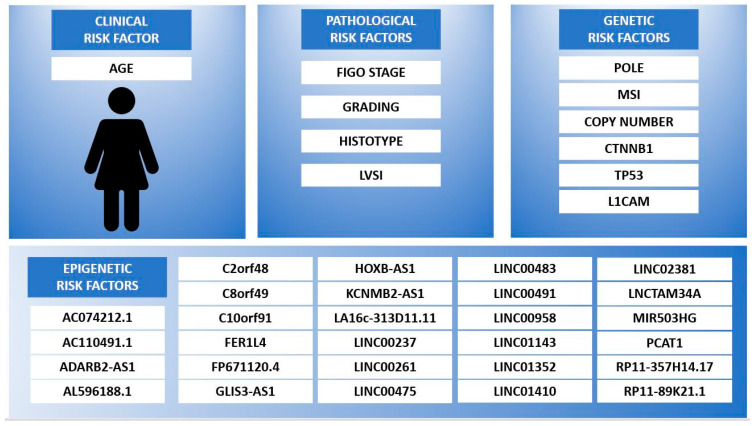
EC patients risk stratification according to clinical, pathological, genetic and epigenetic prognostic factors.

**Table 1 healthcare-09-00965-t001:** Endometrial cancer FIGO staging system.

FIGO Stage	Tumor Spread
Stage I	The tumor is limited to the uterus (it can affect the glands of the cervix and isthmus but not the cervical stroma)
IA	Myometrial invasion <50% of its full thickness
IB	Myometrial invasion >50% of its full thickness
Stage II	The tumor involves the cervical stroma (it does not invade tissues outside the uterus)
Stage III	The tumor has spread outside the uterus (it does not involve the rectum or bladder)
IIIA	Uterine serosa and/or adnexa invasion
IIIB	Vagina and/or parametrial invasion
IIIC	Lymph nodes involvement
IIIC1	Pelvic lymph nodes involvement
IIIC2	Para-aortic lymph nodes involvement
Stage IV	The tumor has spread to rectum, bladder or distant locations
IVA	Rectum and/or bladder invasion
IVB	Distant site metastasis

**Table 2 healthcare-09-00965-t002:** Endometrial cancer ESMO/ESGO/ESTRO classification system.

Risk Category	Risk Factor Status
Low	FIGO Stage I
	Endometrioid histotype
	grade 1–2
	<50% myometrial invasion
	LVSI negative
Intermediate	FIGO Stage I
	Endometrioid histotype
	grade 1–2
	≥50% myometrial invasion
	LVSI negative
High-intermediate	FIGO Stage I
	Endometrioid histotype
	grade 3
	<50% myometrial invasion
	regardless of LVSI status
	OR
	FIGO Stage I
	Endometrioid histotype
	grade 1–2
	LVSI unequivocally positive
	regardless of depth of invasion
High	FIGO Stage I
	Endometrioid histotype
	grade 3
	≥50% myometrial invasion
	regardless of LVSI status
	OR
	FIGO Stage II
	OR
	FIGO Stage III
	Endometrioid histotype
	no residual disease
	OR
	Non-endometrioid histotype
	(serous, clear-cell, undifferentiated carcinoma or carcinosarcoma)
Advanced	FIGO Stage III
	residual disease
	OR
	Stage IVA
Metastatic	FIGO Stage IVB

**Table 3 healthcare-09-00965-t003:** A summary of ncRNAs that play a role as oncogenes in EC pathogenesis and for which a functional characterization is available.

ncRNA Name	Exp.	Function	miR Interactions	Mechanism	Refs.
AL161431.1	up	oncogene	miR-1252-5p	a	[37]
C2orf48	up	oncogene	miR-183	n/a	[38]
CCAT1	up	oncogene	miR-181a-5p	n/a	[39,40]
CCAT2	up	oncogene	miR-216b	a; c	[41]
CDKN2B-AS	up	oncogene	miR-125a-5p	d	[42]
CHL1-AS1	up	oncogene	miR-6076	n/a	[43]
circ_0002577	up	oncogene	miR-197	a; b; e	[44,45]
circ_0061140 *	up	oncogene	miR-149-5p	a	[46]
DANCR	up	oncogene	miR-214	b	[47]
DLEU1	up	oncogene	miR-490	b; e	[48,49]
H19	up	oncogene	miR-20b-5p; miR-124-3p; miR-612	a; b; c; e	[50,51,52,53,54,55]
HOTAIR *	up	oncogene	miR-646	b; f	[56,57,58,59,60,61,62]
HOXB-AS1	up	oncogene	miR-149-3p	a; c	[63]
LINC00483	up	oncogene	miR-183; miR-192	b	[38]
LINC00958	up	oncogene	miR-761	c	[36]
LINC01016	up	oncogene	miR-302a-3p; miR-3130-3p	n/a	[64]
LINC01410	up	oncogene	miR-23c	a; b	[65]
LINC-ROR	up	oncogene	miR-145	n/a	[66]
lnc-OC1	up	oncogene	miR-34a	b	[67]
lncRNA-ATB	up	oncogene	miR-126	a; b; e	[68]
LOXL1-AS1	up	oncogene	miR-28-5p	a; b	[69]
NEAT1	up	oncogene	miR-361; miR-144-3p; miR-146b-5p	a	[70,71,72]
NR2F1-AS1	up	oncogene	miR-363	a; b; c	[73]
PCGEM1	up	oncogene	miR-129-5p	a; c; b	[74]
PVT1 *	up	oncogene	miR-195-5p	a; b	[75,76]
RP11-357H14.17	up	oncogene	miR-24-1-5p; miR-27b; miR-143; miR-204; miR-503; miR-4770	n/a	[77]
RP11-89K21.1	up	oncogene	miR-27b; miR-4770; miR-143; miR-204; miR-125a-5p; miR-125b-5p; miR-139-5p; miR-670-3p	n/a	[77]
SNHG16	up	oncogene	miR-490-3p	a	[78]
SNHG8	up	oncogene	miR-152	a	[79]
TUG1	up	oncogene	miR-34a-5p; miR-299	n/a	[80]

ncRNAs are listed in alphabetical order (column 1) and for each we report its expression in EC compared to control (column 2) (either up- or down-regulated), its oncogenic function (column 3), its functional interactions with target miRNA (column 4) and its role in the development of EC (column 5) as reported in the available literature (column 6). In case of no data available, we report “n/a”. In column 5, the following abbreviations were used: a: cell growth; b: apoptosis; c: cell invasion/migration; d: drug resistance; e: EMT transition; f: other. Data in the table are mostly retrieved from http://www.bio-bigdata.com/lnc2cancer/, accessed on 10 March 2021 and updated according to the most recent (1/2018-onward) data available in PubMed (http://pubmed.ncbi.nlm.nih.gov/, accessed on 10 March 2021). Note that ceRNAs marked with the symbol * can be up- and down-regulated, having both oncogenic and tumor suppressor functions; therefore, they are reported both in Table 3 and Table 4.

**Table 4 healthcare-09-00965-t004:** A summary of ncRNAs that play a role as tumor suppressors in EC pathogenesis and for which a functional characterization is available.

ncRNA Name	Exp.	Function	miR Interactions	Mechanism	Refs.
circ_0061140 *	down	tumor suppressor	miR-149-5p	a	[46]
DCST1-AS1	down	tumor suppressor	miR-92a-3p	c	[81]
GAS5	down	tumor suppressor	miR-103; miR-222-3p	b	[82,83]
HOTAIR *	down	tumor suppressor	miR-646	b; f	[56,57,58,59,60,61,62]
LA16c-313D11.11	down	tumor suppressor	miR-205-5p	a; c	[84]
LINC00261	down	tumor suppressor	miR-27a; miR-96; miR-153; miR-182; miR-183	a; c	[85]
LOC134466	down	tumor suppressor	miR-196a-5p	b	[86]
MALAT1	down	tumor suppressor	miR-200c	c; e	[87]
miR143HG	down	tumor suppressor	miR-125a	b	[88]
MIR22HG	down	tumor suppressor	miR-141-3p	a; b	[89]
NIFK-AS1	down	tumor suppressor	miR-146a	a	[90]
PVT1 *	down	tumor suppressor	miR-195-5p	a; b	[78,79]
RP11-395G23.3	down	tumor suppressor	miR-205-5p	a; c	[91]
SNHG5	down	tumor suppressor	miR-25-3p	a; c	[92]
TUSC7	down	tumor suppressor	miR-23b; miR-616	a; e	[93,94]

ncRNAs are listed in alphabetical order (column 1) and for each we report its expression in EC compared to control (column 2) (either up- or down-regulated), its tumor suppressor function (column 3), its functional interactions with target miRNA (column 4) and its role in the development of EC (column 5) as reported in the available literature (column 6). In case of no data available, we report “n/a”. In column 5, the following abbreviations were used: a: cell growth; b: apoptosis; c: cell invasion/migration; d: drug resistance; e: EMT transition; f: other. Data in the table are mostly retrieved from http://www.bio-bigdata.com/lnc2cancer/, accessed on 10 March 2021 and updated according to the most recent (1/2018-onward) data available in PubMed (http://pubmed.ncbi.nlm.nih.gov/, accessed on 10 March 2021). Note that ncRNAs marked with the symbol * can be up- and down-regulated, having both oncogenic and tumor suppressor functions; therefore, they are reported both in Table 3 and Table 4.

**Table 5 healthcare-09-00965-t005:** A summary of ncRNAs that play a role in EC prognosis.

ncRNA Name	Exp.	Function	Deregulation-Related Prognosis	Author	Year	Refs.
AC074212.1	up	oncogene	poor	Wang, Y.	2020	[95]
AC110491.1 *	n/a down	n/a tumor suppressor	poor good	Ouyang, D. Liu, J.	2019 2019	[26,38]
ADARB2-AS1 ^	n/a n/a	n/a n/a	poor poor	Ouyang, D. Xia, L.	2019 2019	[26,30]
AL596188.1	n/a	n/a	good	Ouyang, D.	2019	[26]
C2orf48 ^	n/a up	n/a oncogene	poor poor	Ouyang, D. Liu, J.	2019 2019	[26,38]
C8orf49	n/a	n/a	poor	Xia, L.	2019	[30]
C10orf91	n/a	n/a	poor	Ouyang, D.	2019	[26]
FER1L4	down	tumor suppressor	poor	Kong, Y.	2018	[82]
FP671120.4	up	oncogene	poor	Wang, Y.	2020	[95]
GLIS3-AS1	n/a	n/a	poor	Ouyang, D.	2019	[26]
HOXB-AS1	up	oncogene	poor	Liu, D.	2020	[63]
KCNMB2-AS1	up	oncogene	good	Tang, H.	2019	[96]
LA16c-313D11.11	down	tumor suppressor	good	Xin, W.	2020	[84]
LINC00237	n/a	n/a	good	Ouyang, D.	2019	[26]
LINC00261 *	up down	oncogene tumor suppressor	good poor	Ouyang, D. Zhao, D.	2019 2019	[26,35]
LINC00475	down	tumor suppressor	good	Tang, H.	2019	[96]
LINC00483 ^	up n/a	oncogene n/a	poor poor	Liu, J. Xia, L.	2019 2019	[30,38]
LINC00491 ^	n/a n/a	n/a n/a	poor poor	Xia, L. Ouyang, D.	2019 2019	[26,30]
LINC00958	up	oncogene	good	Wang, Y.	2019	[36]
LINC01143	up	oncogene	poor	Tang, H.	2019	[96]
LINC01352	down	tumor suppressor	poor	Tang, H.	2019	[96]
LINC01410	up	oncogene	poor	Lu, M.	2020	[65]
LINC02381	up	oncogene	poor	Wang, Y.	2020	[95]
LNCTAM34A	down	tumor suppressor	good	Wang, Y.	2020	[95]
MIR503HG	down	tumor suppressor	poor	Tang, H.	2019	[96]
PCAT1	up	oncogene	poor	Zhao, X.	2019	[97]
RP11-357H14.17	up	oncogene	poor	Gao, L.	2020	[77]
RP11-89K21.1	up	oncogene	poor	Gao, L.	2020	[77]

ncRNAs are listed in alphabetical order (column 1) and for each we report its expression in EC compared to control (column 2) (either up- or down-regulated), its tumor suppressor or oncogene function (column 3), its prognostic value (column 4) and the name of first author (column 5) and year of publication (column 6), as reported in the available literature (column 7). In case of no data available, we report “n/a”. Data in the table are mostly retrieved from PubMed (http://pubmed.ncbi.nlm.nih.gov/, accessed on 10 March 2021). Note that ncRNAs marked with the symbol * are associated to good and poor prognosis in different papers. NcRNAs marked with the symbol ^ have been the subject of multiple studies showing the same prognostic value.

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
