# Peer review of "Towards Personalized Medicine: Non-Coding RNAs and Endometrial Cancer"

_healthcare, 2021, doi:10.3390/healthcare9080965_

Round 1
Reviewer 1 Report
This paper provides a comprehensive summary of the current EC classifications and prognostic role of non-coding RNAs in therapies and cancer relapse. I don't have major comments but would recommend limiting discussion about clinical EC classifications and combine all the clinical classifications into one table. Discussion about EC histology seems very lengthy and in order to be in the scope of the journal I would recommend limiting that to one paragraph.
Reviewer 2 Report
Personalized medicine has been regarded as a major practical outcome of the Human Genome Project. When personalized medicine is incorporated into the clinical setting, it can be treated with the most appropriate medical treatment depending on the genetic and protein profile of the patient. This review article submitted is well consisted and is believed to provide important information to the readers.
On the other hand, it is also true that many similar treatises have been reported. Miki reported a brief and interesting summary of new views, including non-coding of endometrial cancer (Cancers 2020, 12(9), 2595). The review article by Dwivedi SKD et al. is a very novel and unique report (Cancers 2021, 13(5), 1085). The authors also reported a similar review in 2021 (Ref#8). The authors should clarify the difference between the reviews already reported and this submitted review, and then clarify the novelty of your review.
